# FedQLoRA: Federated Quantization-Aware LoRA for Large Language Models

## Abstract

Large language models (LLMs) with billions of parameters have achieved remarkable success across various applications, but they require substantial computational resources and large datasets. While parameter-efficient fine-tuning methods like LoRA and QLoRA have significantly reduced computational costs and memory usage, robustly training LLMs for individual clients with datasets distributed on isolated devices remains challenging. To address this, recent work has explored the use of federated learning (FL) to collaboratively train LLM adapters on distributed private data, thereby avoiding the high computational and communication costs. In these approaches, the LLMs are frozen, and the adapters are collaboratively trained through adapter-sharing and aggregation methods. However, in this paper, we identify a significant issue: these approaches may suffer from quantization bias when clients operate with different levels of quantization on LLMs. To resolve this, we propose a novel framework called **Fed**erated **Q**uantization-Aware **LoRA** (FedQLoRA), which estimates the quantization error and separates it from the LoRA adapter trained on local data via a quantization-aware adapter. Additionally, we address the heterogeneity bias problem that arises from severe data heterogeneity among clients, such as in non-IID settings. We propose an iterative version of the framework that improves both the dynamic quantization-aware adapter and the LoRA adapter alternately within the FL framework. We conduct extensive experiments to validate the performance of our proposed framework. The code and data are available on the website [link].

## 1 Introduction

Large language models (LLMs) have achieved remarkable success across various applications, primarily due to their impressive performance. However, with billions of parameters, fine-tuning these models demands substantial computational resources and large datasets. Parameter-efficient fine-tuning (PEFT) techniques, such as Low-Rank Adaptation (LoRA) Hu et al. (2021), address this issue by freezing most of the pre-trained LLM parameters and updating only a small subset. To further minimize memory usage, learnable quantization methods like QLoRA Dettmers et al. (2024) employ a high-precision quantization approach on pre-trained models while incorporating a small set of learnable low-rank adapter weights. These methods significantly reduce computational costs and memory usage, making LLMs more efficient and adaptable for heterogeneous devices.

Despite their success, PEFT methods like QLoRA/LoRA still require substantial training data to achieve significant improvements over the base model, which may be spread across distributed devices. Collaboratively training LLMs on this distributed data while ensuring privacy protection presents a significant challenge. Recently, many studies have explored applying federated learning (FL) to train LLMs on isolated data across distributed devices. Unlike traditional FL methods, which share all local model parameters with a central server, the massive number of parameters in LLMs (billions) makes this type of communication highly inefficient. To address this, most recent approaches freeze the LLMs and instead train only the adapters, which consist of a much smaller set of parameters. These adapters are trained locally on each client and then aggregated at the server, significantly reducing communication overhead. For example, Cho et al. (2023) proposed a method that aggregates heterogeneous LoRA modules using zero-padding and redistributes them heterogeneously through truncation. Similarly, Sun et al. (2024) addressed synchronization issues with LoRA in FL by introducing FFA-LoRA, which freezes the non-zero-initialized low-rank matrices

and updates only the zero-initialized ones. By focusing on sharing and aggregating adapters across clients, these methods significantly reduce both computational and communication costs.

When applying the adapter-sharing and aggregation method to clients with heterogeneous LLM models using different levels of quantization, we observed an unexpected performance drop, as shown in Figure 1. We conducted experiments with four clients across three scenarios: 1) all clients with LLMs using 2-bit quantization, 2) all clients with LLMs using 4-bit quantization, and 3) mixed LLMs, where two clients used 2-bit quantization and the other two used 4-bit quantization. We implemented one recent approach, FFA-LoRA (Sun et al. (2024)), that follows the adapter-sharing and aggregation framework and tested the impact of two types of quantization, i.e., quantile quantization and LoRA-aware Quantization(Li et al. (2023)). As expected, the 4-bit models outperformed the 2-bit models. However, the mixed-quantization models performed significantly worse than both the fully 2-bit and fully 4-bit models. This unexpected performance drop is caused by a *quantization bias* that arises when aggregating adapters from models with different quantization levels. Specifically, during local training, models attempt to improve performance in two ways: first, by compensating for the loss caused by parameter quantization (which is greater in lower-bit quantization), and second, by enhancing the model's ability to capture information from the local data. Since the quantization loss varies significantly between different quantization levels, aggregating adapters across mixed quantization levels introduces adverse effects, leading to poorer overall performance.

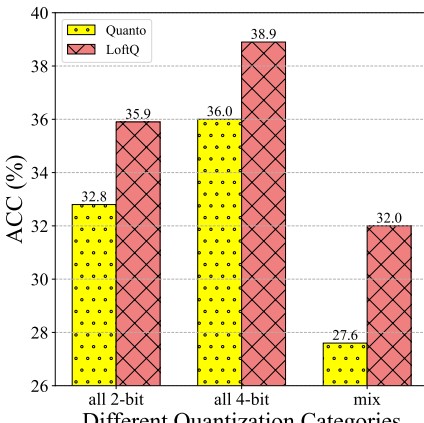

Figure 1: Performance of clients with heterogeneous quantized LLMs. Mixed levels of quantization significantly reduce the performance of FL.

To address the issue of quantization bias, we propose a novel framework called **Fed**erated **Q**uantization-aware **LoRA** (FedQLoRA). This framework is designed to estimate quantization error and separate it from the information learned from local data within the adapter. A key challenge in this approach is that we typically do not have access to the unquantized LLM, making it difficult to measure the quantization error directly. To overcome this, we propose approximating the unquantized model using the locally quantized LLM along with the LoRA adapter trained on local data. The quantization error can be estimated then by calculating the quantization loss of this approximated unquantized model. We formally introduce a *quantization-aware adapter* that compensates for the quantization error in the quantized LLM and reduces the memory usage during inference. The LoRA adapter is then retrained on the local quantized LLM with quantization-aware adapter using local data. This process allows the LoRA adapter to effectively separate quantization errors while capturing unbiased information from local data.

Although a locally quantized LLM paired with a LoRA adapter provides a reasonable approximation of the unquantized model, it may still suffer from *heterogeneity bias* when faced with significant data heterogeneity among clients, such as in non-IID settings. In these cases, each local client might use different sampling methods from the global distribution to generate their heterogeneous local datasets, leading to discrepancies between the models trained on them. To address this, instead of using local data directly, we propose utilizing a LoRA adapter trained on global data to estimate the unquantized LLMs better. We introduce an iterative version of FedQLoRA, incorporating a dynamic quantization-aware adapter. Specifically, we first train LoRA adapters based on local data and then aggregate these at the server to form a global LoRA adapter. This global adapter is then used to update both the local LoRA adapters and the quantization-aware adapters. The updated quantization-aware adapters provide a more accurate estimation of quantization error. With these improved adapters, we can train the local LoRA adapters using local data while minimizing the impact of quantization bias, initiating a new iteration. During this process, the quantization-aware adapter becomes more accurate through the improved LoRA adapter, and the LoRA adapter, in turn, is refined by the more precise quantization-aware adapter.

**In summary**, we have the following contribution: 1) We are the first to identify the issue of quantization bias when applying the adapter-sharing and aggregation method to clients with heterogeneous

LLMs that use different quantization levels. 2) We propose a novel framework, FedQLoRA, which can separate the quantization error from the LoRA adapter through the use of a quantization-aware adapter. Additionally, we introduce an iterative version that addresses heterogeneity bias by alternately improving the dynamic quantization-aware adapter and the LoRA adapter within the FL framework. 3) We conduct extensive experiments to validate the effectiveness and superior performance of our proposed framework.

## 2 BACKGROUND

### 2.1 PRELIMINARIES

**Quantization & Dequantization:** Given a high-precision number $X \in \mathbb{R}$, the quantization process converts it into an integer $X^K \in \{0, 1, \cdots, 2^K - 1\}$, using fewer bits for storage through a quantization encoder $\mathcal{Q}$. This process is defined as

$$X^K = \mathcal{Q}(X) = round\big((2^K - 1)F(X)\big), \tag{1}$$

where $F(\cdot) : \mathbb{R} \rightarrow [0, 1]$ is a normalization function designed based on the input distribution. For example, in uniform quantization, the normalization function is given by $F(X) = (X - X_{\min})/(X_{\max} - X_{\min})$, assuming a uniform distribution. The quantized value must be dequantized through a dequantization decoder before use. This dequantization process involves a lookup table that maps the integer $X^K$ back to its high-precision counterpart $\tilde{X} \in \mathbb{R}$, shown as follows:

$$\tilde{X} = \mathcal{D}\big(X^K\big) = F^{-1}\Big(\frac{X^K}{2^K - 1}\Big), \tag{2}$$

where $F^{-1}(\cdot) : [0, 1] \rightarrow \mathbb{R}$ is an inverse function of $F(\cdot)$.

**Low-rank Adapter:** Low-rank Adapter (LoRA) fine-tuning(Hu et al. (2021)) reduces memory requirements by introducing a small set of trainable parameters, called adapters, while keeping the full model parameters fixed. The core idea behind LoRA is to reduce the number of trainable parameters by decomposing the weight update matrix into two low-rank matrices. Specifically, the trainable low-rank decomposed matrix is defined as $\Delta W \in \mathbb{R}^{d \times l}$, constructed by the product $\Delta W = BA$ where $B \in \mathbb{R}^{d \times r}$ and $A \in \mathbb{R}^{r \times l}$, with $r \ll \min(d, l)$. The total number of parameters is reduced by a factor of $O(r/\min(d, l))$, compared to full fine-tuning of the original weight matrix $W$.

**LoRA-aware Quantization:** LoRA-aware Quantization, as proposed by Li et al. (2023), is designed for pre-trained models that require both quantization and LoRA fine-tuning. This approach integrates low-rank approximation with quantization to jointly approximate the original high-precision pre-trained weights $W \in \mathbb{R}^{d_1 \times d_2}$. Specifically, we define a $K$-bit quantized weight matrix $Q \in \mathbb{R}^{d_1 \times d_2}$ along with low-rank approximations $A \in \mathbb{R}^{d_1 \times r}$ and $B \in \mathbb{R}^{d_2 \times r}$. The LoRA-aware quantization process is then performed by minimizing the following objective:

$$\min_{Q,B,A} |W - Q - BA|. \tag{3}$$

### 2.2 PROBLEM DEFINITION

Federated learning enables multiple clients to train a global model collaboratively without exposing their private data. Suppose we have $N$ clients, each with locally collected data that is inaccessible to others. Each client $i$ has its own private dataset $D_i$ and a personalized large language model (LLM) with parameters $W_i^{q_i}$, which may be quantized to lower precision to reduce memory usage. The quantized parameters $W_i^{q_i}$ for the LLMs may have different levels of quantization denoted by $q_i$ and originate from different unquantized LLMs, but all share the same model structure. This ensures that the adapters at each client are of the same size and can be aggregated directly. Unlike traditional FL methods, which typically share all local model parameters with a central server, the massive number of parameters in LLMs (often in the billions) makes this type of communication highly inefficient. In this work, we keep the quantized LLMs fixed and focus on the adapters, which consist of a small set of parameters. Recent FL research has established a consensus that the knowledge acquired by individual clients consists of both general knowledge and client-specific knowledge. In this paper, we adopt a personalized federated learning framework, where we consider a shared component $\sigma$

to learn general knowledge and a personalized component $\tau_i$ to capture client-specific knowledge. Formally, the training objective can be formulated as follows:

$$\text{Server: } \min_{\sigma,\{\tau_i\}} \sum_{i=1}^{N} \frac{|D_i|}{|D|} \mathcal{L}_i(\sigma, \tau_i; W_i^{q_i}, D_i), \text{ Client i: } \min_{\sigma,\tau_i} \mathcal{L}_i(\sigma, \tau_i; W_i^{q_i}, D_i), \tag{4}$$

where $D = \{D_i\}_{i=1}^{N}$ represents the dataset comprising $N$ heterogeneous local datasets, each of size $|D_i|$. Training is carried out over $R$ rounds. In each round, the server distributes the aggregated, updated LoRA parameters to all clients for further updates.

## 3 METHOD

### 3.1 FRAMEWORK OVERVIEW

In this subsection, we introduce an overview of the FedQLoRA framework. As illustrated in Figure 2, each client consists of three main components: the quantized LLM, the quantization-aware adapter, and the LoRA adapter. The quantized LLM remains frozen throughout the process, while the two adapters are trained alternately based on OPT1 in (19) and OPT2 in (20). First, the quantization-aware adapter is optimized using OPT1 while keeping the LoRA adapter frozen. Afterward, the LoRA adapter is trained from local data, using both the quantized LLM and the frozen quantization-aware adapter based on OPT2. Once training is complete, the LoRA adapter is sent to the server, which aggregates the LoRA adapters from all clients. The aggregated LoRA adapter is then distributed back to each client, where it updates the LoRA adapter in both the OPT1 and OPT2 stages. The quantization-aware adapter then resumes optimization, beginning a new iteration. After federated learning (FL) training is complete, each client proceeds to inference using the quantized LLM and both adapters. The quantization-aware adapter compensates for client-specific quantization errors in the LLM, while the LoRA adapter leverages distributed data cooperatively to fine-tune the LLM, without compromising the privacy of datasets across different clients.

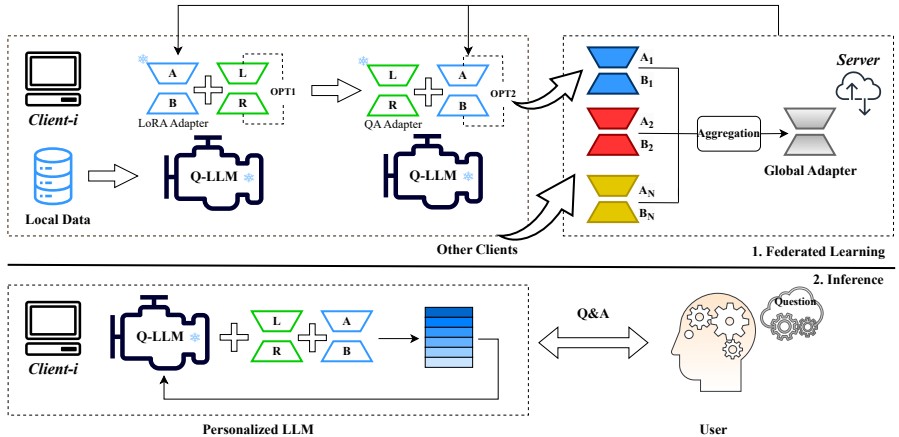

Figure 2: Framework overview of FedQLoRA

### 3.2 FEDERATED QUANTIZATION-AWARE LORA

**Quantization bias:** For client $i$, directly decoding the quantized model using $W_i^{q_i}$ to simulate a high-precision model $\tilde{W}_i$ would lead to significant performance degradation. QLoRA is a widely used method that trains a low-rank adapter for the quantized model based on local data, helping to mitigate the potential loss caused by quantization. Additionally, the adapter trained on local data can capture the unique characteristics of each individual client. When using QLoRA to update the low-rank adapter, we obtain:

$$Y^F = X^F \mathcal{D}(W_i^{q_i}) + X^F B_i A_i, \tag{5}$$

where $\mathcal{D}(\cdot)$ is the quantization decoder. QLoRA dequantizes the stored data type (e.g., q-bit quantized integers) into the computation data type (e.g., full-precision floats) and computes weight gradients for the LoRA parameters using the computation data type.

When using the unquantized model (e.g., full precision) to train the adapter that learns the personalized characteristics of client $i$, the adapter becomes more accurate because it avoids any quantization loss. Consequently, the adapter is able to more precisely capture the true characteristics of the local data for each client. By using LoRA to update the unquantized model, we have

$$Y^F = X^F W_i + X^F B_i^* A_i^*. \tag{6}$$

When comparing the adapters of these two methods, we find that the difference between equations 5 and 6 must satisfy the following condition:

$$\mathbb{E}_{X^F}\left[X^F\left(B_i A_i - B_i^* A_i^* - (W_i - \mathcal{D}(W_i^{q_i}))\right)\right] \equiv 0. \tag{7}$$

Then, the LoRA adapter will bring a *quantization error* after quantization as follows:

$$E_i \triangleq B_i A_i - B_i^* A_i^* = W_i - \mathcal{D}(W_i^{q_i}). \tag{8}$$

Note that the quantization error is endogenous and determined by three factors: 1) the unquantized model $W_i$, 2) the quantization precision $q$, and 3) the quantization method for $W_i^{q_i}$. When any of these factors differs across clients, the resulting quantization error will be heterogeneous.

When applying federated learning to aggregate information from different clients, the difference between the adapter of the unquantized model and that of the quantized model will change. To illustrate this, we analyze the typical FedAVG method. In FedAVG, after each client updates its adapter, the aggregated local adapter becomes $\frac{1}{N}\sum_i B_i A_i$ for the quantized model, and $\frac{1}{N}\sum_i B_i^* A_i^*$ for the unquantized model. We can break down their difference into two components, as follows:

$$\frac{1}{N}\sum_j B_j A_j - \frac{1}{N}\sum_j B_j^* A_j^* = \underbrace{\frac{1}{N}\sum_j (E_j - E_i)}_{\text{Quantization bias}} + \underbrace{E_i}_{\text{Error}} \tag{9}$$

Since the quantization error for client $i$ is endogenous, it remains unchanged during the FL process and contributes to part of the overall gap. However, when we exclude the quantization error from this gap, we still observe an additional error. We refer to this as *quantization bias*, which represents the average difference in quantization errors between client $i$ and the other clients. When clients have homogeneous quantization errors, the quantization bias should be zero. The quantization bias explains the performance degradation observed when aggregating adapters from clients with models that use different levels of quantization, as shown in Fig. 1. Due to the presence of quantization bias, directly aggregating adapters on the server can negatively impact the learning process.

To address this issue, we propose estimating the quantization error and separating it from the LoRA adapter. By aggregating the LoRA adapter after removing the quantization errors, the quantization bias is effectively eliminated.

**Quantization-aware Learning:** To allow the adapter to accurately capture the characteristics of the local data, it is preferable to separate the quantization error. First, let's consider the ideal scenario where each client has access to the unquantized model. However, given the billions of parameters and the enormous memory requirements, it is more practical to use a low-rank adapter to learn the quantization error. When using the quantized model, the adapter can compensate for the quantization loss. The optimal low-rank adapter can be learned by solving the following optimization problem:

$$\min_{L_i, R_i} |W_i - \mathcal{D}(W_i^{q_i}) - L_i R_i|. \tag{10}$$

The learned optimal matrices $L_i, R_i$ can compensate for the error caused by quantization loss. To solve this optimization problem, we apply Singular Value Decomposition (SVD) to estimate the quantization error. We define the rank of the quantization error matrix $E$ as $m$ where $0 < m < \min\{d_1, d_2\}$, then the quantization error can be expressed as

$$E_i = \sum_{s=1}^{d} \sigma_i \mathbf{u}_i \mathbf{v}_i^T \approx \sum_{s=1}^{m} \sigma_i \mathbf{u}_i \mathbf{v}_i^T \tag{11}$$

where $d = min\{d_1, d_2\}, \sigma_1 \geq \sigma_2 \geq \cdots \geq \sigma_d$ are the singular values, $\mathbf{u}_i$ and $\mathbf{v}_i$ are the associated left and right singular vectors of $E_i$. We then obtain a rank-$m$ adapter $L, R$ by

$$L = [\sqrt{\sigma_1}u_1, \cdots, \sqrt{\sigma_m}u_m], R = [\sqrt{\sigma_1}v_1, \cdots, \sqrt{\sigma_m}v_m] \tag{12}$$

**Proposition 1** *If $W_i^{q_i}$ is quantized from $W_i$ via LoRA-aware quantization in (3), i.e., $W_i^{q_i}, \hat{A}_i, \hat{B}_i = \mathcal{Q}(W_i)$, then the optimal quantization-aware adapter in (10) satisfies $R_i^* = \hat{A}_i$ and $L_i^* = \hat{B}_i$.*

This proposition demonstrates the relationship between LoRA-aware quantization and the quantization-aware adapter. In practice, the quantization applied to different clients may vary depending on the type of application.

**Federated Quantization-aware Learning:** Previous quantization-aware learning approaches assume that each client stores both the quantized model and the full-precision unquantized model, which is often not the case in reality. Typically, clients only store the quantized LLM and may not have access to the unquantized model due to either commercial or technical constraints. To estimate the quantization error, we aim to approximate the unquantized model using the local data. Specifically, we train a LoRA adapter with parameter $\tilde{B}_i$ and $\tilde{A}_i$ for quantized model based on the local data $D_i$ according to Eq. 5. Based on the LoRA adapters, we estimate the unquantized model as:

$$\tilde{W}_i = \mathcal{D}(W_i^{q_i}) + \tilde{B}_i\tilde{A}_i. \tag{13}$$

Based on the approximated full-precision unquantized model $\tilde{W}_i$, we can re-quantize it to generate the quantized model through the quantization encoder $\mathcal{Q}_i$. This allows us to estimate the quantization error using quantization-aware learning. The optimization then aims to minimize the quantization error with the following objective:

$$\min_{L_i, R_i} \left| \tilde{W}_i - \mathcal{D}_i(\mathcal{Q}_i(\tilde{W}_i)) - L_iR_i \right|. \tag{14}$$

When the quantization method for the quantized model $W_i^{q_i}$ is known, we can apply the same quantization method $\mathcal{Q} = \mathcal{Q}_i$. This allows us to eliminate the term $W_i^{q_i}$ since $\mathcal{D}(\mathcal{Q}(\mathcal{D}(W_i^{q_i}))) = \mathcal{D}(W_i^{q_i})$. As a result, the above optimization in (14) can be simplified as follows:

$$\min_{L_i, R_i} \left| \tilde{B}_i\tilde{A}_i - \mathcal{D}(\mathcal{Q}(\tilde{B}_i\tilde{A}_i)) - L_iR_i \right|. \tag{15}$$

Based on the optimization, we can obtain the quantization-aware adapter $L_i, R_i$, which compensates for the quantization error caused by quantization loss. Next, we perform additional LoRA adapter training using the quantized model with the quantization-aware adapter, i.e., $\mathcal{D}(W_i^{q_i}) + L_iR_i$. The training of the unbiased LoRA adapter for client $i$ can be expressed as follow:

$$Y^F = X^F\mathcal{D}(W_i^{q_i}) + X^FL_iR_i + X^FB_iA_i. \tag{16}$$

Note that when client i update the LoRA adapters $B_i, A_i$, the quantization-aware adapters $L_i, R_i$ should remain fixed. In a federated learning setting, the LoRA adapter $B_i, A_i$ can be aggregated into the server. If we adopt the FedAVG to aggregate LoRA adapters, the server updates them as:

$$B = \sum_{i=1}^{N} \frac{|D_i|}{|D|} B_i, \quad A = \sum_{i=1}^{N} \frac{|D_i|}{|D|} A_i. \tag{17}$$

The shared parameter is $\sigma = \{B, A\}$, while the personalized parameter is $\tau_i = \{L_i, R_i\}, i = 1, 2, \cdots, N$. The shared parameter would be updated based on Eq.16 and Eq.17 until convergence is reached. Meanwhile, the personalized parameter remains unchanged throughout the federated learning process.

### 3.3 ITERATIVE OPTIMIZATION

In the federated quantization-aware learning approach, local data $D_i$ is used to train the quantization-aware adapter, allowing it to approximate the unquantized model and estimate the quantization error. While a locally quantized LLM paired with a LoRA adapter offers a reasonable approximation, it may still be affected by *heterogeneity bias* in cases where there is significant data heterogeneity

among clients, such as in non-IID settings. The ideal way to avoid heterogeneity bias would be to use the entire dataset $\{D_i\}_{i=1}^N$ to train the LoRA adapter and accurately estimate the quantization error. However, due to data privacy constraints, each client cannot access the complete dataset.

In this subsection, we propose an iterative optimization method that uses a dynamic quantization-aware adapter to gradually capture the quantization error. We define the dynamic quantization-aware adapters as $L_{t,i}, R_{t,i}$, initializing them as zeros. The dynamic LoRA adapters $B_{t,i}, A_{t,i}$ are initializd as $B_{0,i} = \tilde{B}_i$ and $A_{0,i} = \tilde{A}_i$. Both the quantization-aware adapters and the LoRA adapters are incorporated into the estimation of the full-precision unquantized model as follows:

$$\tilde{W}_{t,i} = \mathcal{D}(W_i^{q_i}) + L_{t,i}R_{t,i} + B_{t,i}A_{t,i}. \tag{18}$$

Similar to previous federated quantization-aware learning methods, we can eliminate the term $W_i^{q_i}$ when the quantization method is known. This leads to the following simplified optimization for learning the quantization-aware adapter:

$$\min_{L_{t+1,i}, R_{t+1,i}} |L_{t,i}R_{t,i} + B_{t,i}A_{t,i} - \mathcal{D}(\mathcal{Q}(L_tR_t + B_{t,i}A_{t,i})) - L_{t+1,i}R_{t+1,i}|. \tag{19}$$

Based on the optimization, we can update the quantization-aware adapter $L_{t+1}, R_{t+1}$, which compensates for the quantization loss due to quantization. The LoRA adapter in Eq.18 should also be updated based on the quantized model with new quantization-aware adapter, i.e., $\mathcal{D}(W_i^{q_i}) + L_{t+1,i}R_{t+1,i}$. The training process for the LoRA adapter for client $i$ can be expressed as follows:

$$Y^F = X^F\mathcal{D}(W_i^{q_i}) + X^FL_{t+1,i}R_{t+1,i} + X^FB_{t,i}A_{t,i}. \tag{20}$$

Similarly, when client i update the LoRA adapters $B_{t,i}, A_{t,i}$, the quantization-aware adapters $L_{t+1,i}, R_{t+1,i}$ should remain fixed. In a federated learning setting, when we adopt the FedAVG to aggregate the LoRA adapters $B_{t,i}, A_{t,i}$, the server updates them as follows:

$$B_{t+1} = \sum_{i=1}^N \frac{|D_i|}{|D|}B_{t,i}, \quad A_{t+1} = \sum_{i=1}^N \frac{|D_i|}{|D|}A_{t,i}. \tag{21}$$

The global adapters $B_{t+1}, A_{t+1}$ are then sent back to the clients to update the local LoRA adapters, i.e., $A_{t+1,i} = A_{t+1}$ and $B_{t+1,i} = B_{t+1}$. The shared parameter is $\sigma = \{B_t, A_t\}$ and the personalized parameter is $\tau_i = \{L_{t,i}, R_{t,i}\}, i = 1, 2, \cdots, N$ across all the iterations $t$ before convergence. The client would update the personalized parameter $\tau_i$ via Eq.19 and the shared parameter $\sigma$ via Eq.20, and the server would aggregate the shared parameter by Eq.21 until convergence is achieved.

# 4 EXPERIMENTS

In this section, we first analyze the model's overall performance with homogeneous and heterogeneous data separation across clients. We then provide an in-depth analysis, including the impact of heterogeneous models and datasets, and the convergence analysis.

## 4.1 EXPERIMENTAL SETTING

**Dataset:** We consider two datasets in our experiments, text classification using the 20_Newsgroups dataset Lang (1995) and multi-class news classification using the NC (News Classification) dataset Lewis et al. (2019) from XGLUE Liang et al. (2020). Accuracy and micro-F1 score are used as the evaluation metric for this multi-class classification task. Both datasets are randomly partitioned for training and evaluation, with each client working on a subset of the categories, ensuring a balanced distribution of data across the clients.

**Baseline:** We implement four baselines that adopt the LoRA adapters for LLMs under FL, including "LoRA" with local adapter training based on local data, FFA-LoRA Sun et al. (2024) with aggregation of non-zero LoRA adapter, H-LoRA Cho et al. (2023) with padding and truncation over LLMs and the H-LoRA-T with truncation of LLMs into lowest precision based on H-LoRA.

**Implementation details:** For our experiments, we use the DistilBERT-base-multilingual-cased architecture Sanh (2019) as the LLM backbone. This model is a distilled version of BERT, retaining

97% of BERT's performance while being 60% smaller and faster. We apply LoRA modules specifically to the `q_lin` and `v_lin` layers within the self-attention mechanism. Two levels of quantization are considered: 2-bit and 4-bit quantization. The ratio of these quantization levels is set to 1:1 across all clients. We experiment with three different FL settings, with N = 3, 5, 10 clients. Non-IID data partitions are created using a Dirichlet distribution-based approach with a scale parameter of 1 by default, while IID data is evenly distributed across all clients for comparison. Each local model is evaluated on a globally balanced test dataset. We use the SGD optimizer with a learning rate of 0.001, and all experiments are conducted on an NVIDIA A100 GPU with 80GB of memory.

## 4.2 OVERALL PERFORMANCE

Table 1 demonstrates that our proposed methods, FedQLoRA and its iterative version iFedQLoRA, outperform the baselines in the IID scenario. Since there is no data heterogeneity bias, FedQLoRA performs almost the same as the iterative version. It is worth noting that FedQLoRA conducts quantization-aware optimization only once, resulting in significantly lower computational costs. Additionally, we observe that H-LoRA-T, which truncates LLMs from high precision to the lowest precision, effectively avoids quantization bias, leading to better performance than H-LoRA. However, this method cannot fully utilize the potential of high-precision models. Furthermore, we find that FFA-LoRA does not perform well and is sensitive to hyperparameters.

Table 2 demonstrates that iterative FedQLoRA outperforms both FedQLoRA and the baselines in the non-IID scenario. The iterative FedQLoRA mitigates data heterogeneity bias by learning the quantization-aware adapter based on global data. As the number of clients increases, the advantage of our method becomes more pronounced. This is because the quantization bias has a greater impact on LoRA's learning when each client has fewer data samples. Even in the absence of quantization bias, our method significantly outperforms H-LoRA-T, indicating that any information loss from LLMs is amplified when the data sample size is reduced.

Table 1: Overall performance under IID dataset

| Dataset | XGLUE NC | | | | | | 20 NewsGroup | | | | | |
|---|---|---|---|---|---|---|---|---|---|---|---|---|
| #Clients | 3 clients | | 5 clients | | 10 clients | | 3 clients | | 5 clients | | 10 clients | |
| Method | Acc | F1 | Acc | F1 | Acc | F1 | Acc | F1 | Acc | F1 | Acc | F1 |
| LoRA | 76.7 | 0.630 | 74.7 | 0.586 | 73.1 | 0.569 | 48.0 | 0.463 | 45.4 | **0.437** | 41.9 | **0.410** |
| FFA-LoRA | 44.3 | 0.110 | 40.5 | 0.082 | 9.3 | 0.019 | 5.9 | 0.018 | 6.5 | 0.016 | 5.8 | 0.022 |
| H-LoRA | 78.2 | 0.559 | 76.0 | 0.629 | 74.6 | 0.543 | 47.6 | 0.462 | 39.5 | 0.324 | 35.5 | 0.287 |
| H-LoRA-T | 79.2 | 0.571 | 78.9 | 0.675 | 75.3 | 0.552 | 48.7 | 0.471 | 43.2 | 0.366 | 37.4 | 0.288 |
| FedQLoRA | 80.3 | 0.705 | **79.6** | **0.676** | 77.7 | 0.572 | 48.0 | 0.468 | 45.2 | 0.388 | 42.9 | 0.347 |
| iFedQLoRA | **81.3** | **0.711** | 79.3 | 0.674 | **78.1** | **0.573** | **49.3** | **0.479** | **46.2** | 0.405 | **43.3** | 0.379 |

Table 2: Overall performance under Non-IID dataset

| Dataset | XGLUE NC | | | | | | 20 NewsGroup | | | | | |
|---|---|---|---|---|---|---|---|---|---|---|---|---|
| #Clients | 3 clients | | 5 clients | | 10 clients | | 3 clients | | 5 clients | | 10 clients | |
| Method | Acc | F1 | Acc | F1 | Acc | F1 | Acc | F1 | Acc | F1 | Acc | F1 |
| LoRA | 71.9 | 0.480 | 61.3 | 0.411 | 54.4 | 0.281 | 41.4 | 0.326 | 38.0 | 0.285 | 34.4 | **0.264** |
| FFA-LoRA | 42.9 | 0.115 | 38.9 | 0.056 | 11.4 | 0.047 | 7.4 | 0.020 | 6.4 | 0.027 | 5.6 | 0.022 |
| H-LoRA | 75.4 | 0.547 | 62.6 | 0.420 | 56.6 | 0.252 | 43.8 | 0.415 | 38.8 | 0.332 | 24.8 | 0.160 |
| H-LoRA-T | 76.1 | 0.550 | 63.0 | 0.432 | 58.4 | 0.270 | 45.0 | 0.431 | 41.1 | 0.344 | 29.4 | 0.201 |
| FedQLoRA | 75.8 | 0.553 | 62.8 | 0.403 | 59.0 | 0.250 | 45.4 | 0.433 | 43.0 | **0.416** | 32.7 | 0.219 |
| iFedQLoRA | **77.7** | **0.588** | **63.3** | **0.444** | **61.6** | **0.287** | **46.8** | **0.446** | **45.5** | 0.378 | **34.9** | 0.238 |

## 4.3 IN-DEPTH ANALYSIS

**Model heterogeneity analysis:** As shown in Figure 3, two subgraphs present results for homogeneous and heterogeneous data divisions under various quantization configurations. We first observe that our methods demonstrate relatively robust performance against model heterogeneity in the IID

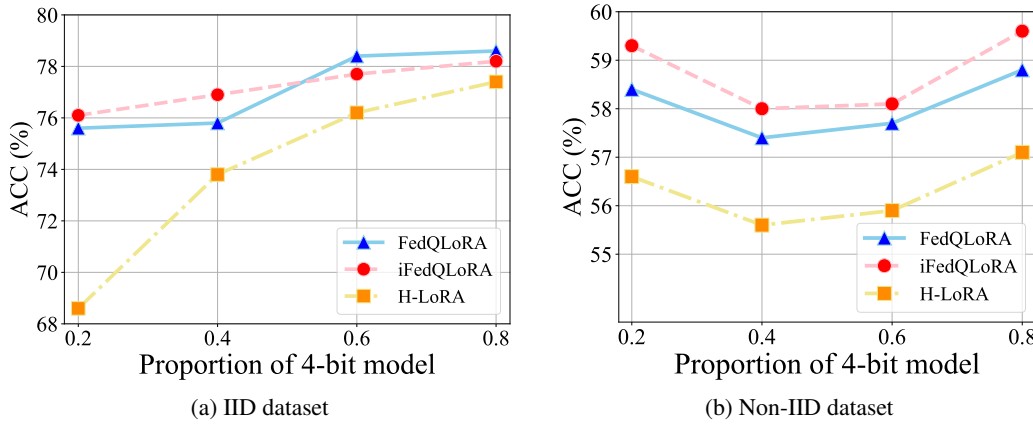

(a) IID dataset

(b) Non-IID dataset

Figure 3: Impacts of model heterogeneity

scenario. For example, our FedQLoRA experiences only a 2% accuracy reduction when the proportion of 4-bit models decreases from 80% to 20%, while H-LoRA suffers almost a 9% accuracy reduction when only 20% of models are 4-bit. In the non-IID scenarios, our proposed methods show consistent improvements over H-LoRA, largely due to the reduction of quantization bias. For instance, iFedQLoRA achieves almost a 2.5% improvement over H-LoRA under varying levels of model heterogeneity. Additionally, the iterative version, iFedQLoRA, consistently outperforms FedQLoRA by mitigating the data heterogeneity bias when training quantization-aware adapters.

**Data heterogeneity analysis:** We use a Dirichlet distribution $G \sim D(\beta, G_0)$ to allocate data among clients, introducing data heterogeneity, where $\beta$ is the scaling parameter and $G_0$ is the base distribution. A larger $\beta$ results in less heterogeneity across the clients' data. The impact of data heterogeneity on the XGLUE NC dataset is analyzed in Fig. 4. As data heterogeneity increases (i.e., as $\beta$ decreases), the performance of all methods declines. However, our proposed method shows greater resilience to data heterogeneity. For instance, iFedQLoRA experiences a 9% reduction in accuracy (ACC), compared to a 15% reduction for H-LoRA when $\beta$ decreases from 2 to 0.5. Additionally, the iterative version, iFedQLoRA, consistently outperforms FedQLoRA under varying degrees of heterogeneity. Notably, the performance improvement of iFedQLoRA over FedQLoRA increases from 0.5% to

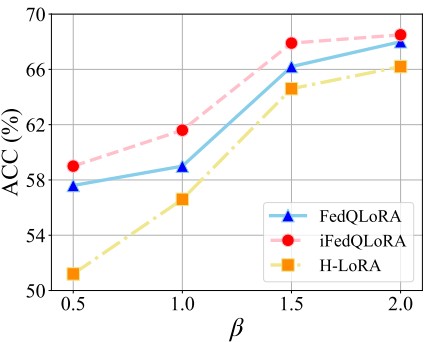

Figure 4: Impacts of data heterogeneity

2.6% as $\beta$ decreases from 2 to 1. This demonstrates the effectiveness of leveraging global data to estimate the quantization-aware adapter.

**Convergence analysis:** As shown in Figure 5, we present the convergence analysis for the XGLUE NC dataset under non-IID data partition settings over forty communication rounds. Our results show that the iterative version, iFedQLoRA, achieves a significantly faster convergence rate compared to H-LoRA. Specifically, iFedQLoRA nearly converges after just 10 communication rounds, while H-LoRA requires at least 20 rounds. Additionally, iFedQLoRA converges slightly faster than FedQLoRA, reaching convergence 5 rounds earlier. Despite introducing the dynamic quantization-aware adapter, iFedQLoRA maintains strong convergence, benefiting from the enhanced performance driven by the improved accuracy of the quantization-aware adapter. Finally, we observe that

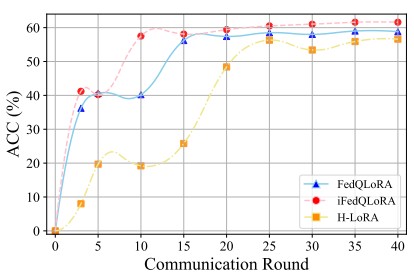

Figure 5: Convergence comparisons

FedQLoRA outperforms H-LoRA in convergence speed, suggesting that isolating the quantization error can enhance the training efficiency of LoRA adapters.

## 5 RELATED WORK

**Parameter efficient fine-tuning:** Houlsby et al. (2019) proposed **Adapters**, which inserts small, learnable layers into pre-trained models. This method allows fine-tuning with fewer parameters, reducing computational demands while maintaining task performance. Lester et al. (2021) demonstrated that prompt tuning scales well with large language models, offering a competitive alternative to full fine-tuning by optimizing continuous prompts with fewer parameters. Li & Liang (2021) introduced **Prefix-Tuning**, which adds learnable prefix vectors to Transformer layers, allowing efficient fine-tuning of generative models by updating only a small set of parameters. Similarly, Zaken et al. (2021) proposed **BitFit**, a method that fine-tunes only the bias terms of Transformer models, further minimizing the number of trainable parameters. Hu et al. (2021) introduced **LoRA (Low-Rank Adaptation)**, which modifies model weights with low-rank matrices during fine-tuning. Expanding on adapter methods, Pfeiffer et al. (2020) proposed **AdapterFusion**, which improves model generalization by combining task-specific adapters during inference. To further reduce parameter updates, Liu et al. (2022) developed **IA3**, which scales activations in self-attention modules, allowing efficient fine-tuning without modifying the core model weights. Dettmers et al. (2024) introduced **QLoRA**, which combines LoRA with quantization, reducing memory and computational costs for fine-tuning large models on resource-limited devices. Additionally, Li et al. (2023) proposed **LoftQ**, a novel quantization method that optimizes LoRA and quantized backbone weights together, improving LoRA fine-tuning in low-bit quantization scenarios.

**Federated efficient fine-tuning:** Recently, several preliminary papers have emerged on federated learning for large language models (LLMs), focusing on updating and sharing adapters in parameter-efficient fine-tuning (PEFT). FederatedScope-LLM(Kuang et al. (2024)) explored the overall federated architecture in LLMs using various PEFT methods, while OpenFedLLM(Ye et al. (2024)) built an integrated framework/codebase that supports federated instruction tuning. Among all PEFT methods, the LoRA adapter significantly reduces communication costs while demonstrating excellent performance in federated learning. Some studies have specifically examined federated LoRA adapters in the context of data heterogeneity. For instance, FFA-LoRA(Sun et al. (2024)) addressed synchronization issues in federated learning by freezing the non-zero-initialized low-rank matrices and updating only the zero-initialized ones. Other works have focused on model heterogeneity. One study(Cho et al. (2023)) proposed a method that aggregates heterogeneous LoRA modules using zero-padding and redistributes them heterogeneously through truncation. Additionally, FDLoRA(Qi et al. (2024)) introduced a variant of personalized federated learning that utilizes dual LoRA tuning. Another framework, pFedLoRA(Yi et al. (2024)), was designed for model-heterogeneous personalized federated learning based on LoRA tuning and included an iterative training method that alternates between training homogeneous small adapters and heterogeneous LLMs. Our proposed methods focus on federated learning across clients with large language models (LLMs) that have varying levels of quantization, and it is the first work to identify and address quantization bias.

## 6 CONCLUSION

In this work, we explored federated fine-tuning with adapters for clients using heterogeneous LLMs with varying levels of quantization. We are the first to identify the issue of quantization bias when applying the adapter-sharing and aggregation method in FL, as adopted by most recent work. To address this, we proposed a novel framework called FedQLoRA, which estimates and separates quantization error from the LoRA adapter trained on local data through the use of a quantization-aware adapter. Additionally, we tackled the issue of heterogeneity bias that arises from significant data heterogeneity among clients, such as in non-IID settings. We introduced an iterative version of the framework that alternately improves both the dynamic quantization-aware adapter and the LoRA adapter within the FL framework. We would explore the application of FedQLoRA to more real-world industrial scenarios, such as recommendation systems, in our future work.

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
