# OpenReview forum: "FedQLoRA: Federated Quantization-Aware LoRA for Large Language Models"
_ICLR.cc/2025/Conference — Submitted to ICLR 2025_

### Official Review · Reviewer_shaa · 2024-10-29

**Soundness:** 2
**Presentation:** 3
**Contribution:** 1
**Rating:** 3
**Confidence:** 5

**Summary:**

This paper addresses the challenge of efficiently training large language models (LLMs) in federated learning (FL) environments, where datasets are distributed across isolated devices. To address quantization bias and data heterogeneity, the authors introduce Federated Quantization-Aware LoRA (FedQLoRA), a framework that mitigates quantization and heterogeneity biases through a quantization-aware adapter and iterative training process. Extensive experiments demonstrate the effectiveness of FedQLoRA, with code and data provided for transparency and reproducibility.

**Strengths:**

1) The authors provided code and data for transparency and reproducibility.
2) The writing is easy to understand.

**Weaknesses:**

1) I believe the quantization bias proposed by the authors is uncommon in real-world scenarios, as the quantization method can be pre-defined across clients. This seems more like a contrived scenario.
2) This scenario of quantization bias could only arise due to differences in computational resource across clients, where some clients intentionally choose different quantization methods. As shown in Table 1, “mix” performs even worse than “all 2-bit”. So, why not simply use the 2-bit (lower resource) method directly for all clients? In general, this issue is difficult to understand and does not align with real-world scenarios.
3) The authors repeatedly claim 'LLMs (billions),' but the model actually used (DistilBERT-base-multilingual-cased) is only 134M, which cannot be considered a LLM. And the authors only conduct experiments on natural language understanding tasks. The author should conduct experiments on more popular LLM tasks like natural language generation [1, 2] with larger LLMs like llama-2-7B.
[1] OpenFedLLM: Training Large Language Models on Decentralized Private Data via Federated Learning.
[2] FedLLM-Bench: Realistic Benchmarks for Federated Learning of Large Language Models.
4) FedQLoRA should be compared with some personalized federated learning methods + LoRA.
5) The experiments are insufficient. Additional experiments should be included, such as using different network architectures, partial participation scenarios, various quantization methods (like 8-bit), and more datasets.
6) The proposed method's approach to addressing quantization bias and heterogeneity bias is unclear. The authors should provide more detailed explanations and conduct additional empirical and theoretical analyses.
7) What is the main focus of this paper? Is it quantization bias or heterogeneity bias? These are two completely different issues: one arises from model heterogeneity, while the other stems from data heterogeneity.
8) An analysis of the computational and communication overhead should be included.

**Questions:**

Please see Weakness.

---

### Official Review · Reviewer_MsA2 · 2024-11-02

**Soundness:** 3
**Presentation:** 3
**Contribution:** 3
**Rating:** 5
**Confidence:** 4

**Summary:**

This paper addresses challenges in fine-tuning large language models (LLMs) in distributed environments using federated learning (FL). While LLMs with billions of parameters perform impressively across tasks, they demand significant computational resources, especially when fine-tuning on isolated, distributed devices. Parameter-efficient techniques like LoRA and QLoRA help reduce these costs, but training LLMs robustly across devices with differing quantization levels remains difficult. The authors highlight a quantization bias problem in FL-based fine-tuning methods where clients use varying quantization levels, which can lead to inconsistencies in model updates. To address this, they propose a new framework, Federated Quantization-Aware LoRA (FedQLoRA). This approach estimates and separates the quantization error from the LoRA adapter updates via a dedicated quantization-aware adapter, helping to mitigate quantization bias. Additionally, they tackle the challenge of data heterogeneity across clients, especially in non-IID settings, by introducing an iterative method that dynamically improves both quantization-aware and LoRA adapters. The paper’s experiments demonstrate the effectiveness of this approach in enhancing LLM performance in federated settings.

**Strengths:**

- It is crucial to conduct research for quantization-aware training on LLMs in the FL scenario, as it can bring more accurate yet high-privacy models.
- The paper is well-written and easy to understand.

**Weaknesses:**

- This paper seems a bit A+B for me and just solves a very limited problem in the scope of mixed-precision FL scenario with quantization error compensation, meanwhile, the high-level idea is highly related to the LoftQ (a variant of Qlora), i.e., both are seeking the way to compensate quantization error by SVD. Moreover, in the paper entitled "Large language models", I, therefore, believe the experiments should be conducted on modern LLMs like LLama families.
- Some implementation details are unclear. How do the authors initiate the LoRA modules? According to the Fig.1 of the Introduction, they seem to be initialized using the method of LoftQ (a variant of Qlora), however, the authors claim that the unquantized model is inaccessible in the local node, as far as I know, the initialization process of LoftQ needs to conduct matrix decompositions for the quantization error, which requires the participation of both quantized model and unquantized model. Or does the above process totally run on the central server?
- The authors claim they approximate the quantization error on the quantized LLMs along with the adapter using local data, however, are there any empirical results to show this approximation is correlated with the true quantization error? Since the experiments are conducted with simulations, I think we can simply dump the quantization error to calculate the correlation.
- Writing issue: Line 178 OPT1 in (19) and OPT2 in (20) should be OPT1 in Equ. (19) and OPT2 in Equ. (20).

**Questions:**

Please refer to the weakness.

---

### Official Review · Reviewer_mWVv · 2024-11-04

**Soundness:** 3
**Presentation:** 3
**Contribution:** 3
**Rating:** 5
**Confidence:** 4

**Summary:**

The paper studied federated fine-tuning of large language models with different quantization levels.
The different quantization levels across clients will introduce quantization bias, which affects the model fine-tuning performance. The work proposed a framework called FedQLoRA that can estimate and separate quantization error from the LoRA adaptor trained on local data. The basic idea of FedQLoRA is to train another set of LoRA parameters in parallel to the original LoRA adaptor and alternative their training to improve the training performance. Evaluation using DistillBERT model and 2-bit/4-bit quantized model shows some better converged accuracy compared to baselines.

**Strengths:**

- study a new problem federated learning setting where clients fine-tune LLMs with different quantization levels.
- introduce the concept of quantization bias to explain the poor performance when clients' LLMs are of different quantization level
- promising results compared to compared baselines

**Weaknesses:**

The paper brought an interesting observation: when multiple clients train a LLM with different quantization levels, the quantization bias impacts the converged accuracy. The proposed approach sounds reasonable and straightforward to implement. With that said, the paper could be improved in the following aspects:

- The rationale behind approximating quantization error using the introduced adaptors is not clear to me. Why the adaptor can estimate the quantization error using the local data? Are there any other approaches to approximate the quantization error? Can the paper demonstrate that, assuming the original floating point models are available, did the learnt quantization-aware adaptor really approximate the quantization error at all? I would suggest the author to evaluate how well the introduced adaptor can approximate the quantization error.

- It looks to me that the quantization bias can be treated as a type of data heterogeneity across clients. Here the data refers to not the raw data, but the intermediate features computed from the model. Under this context, the proposed approach seems to be a personalization approach that personalizes the feature extraction process to the local data distribution. If we look at the problem from this perspective, then existing personalization approaches should be compared as baselines. I would suggest the authors justify why prior personalization approaches won't work to address the quantization bias problem.

Some other minor comments:
- Although the work focuses on LLMs, the tasks the proposed approach is evaluated on are simple classification tasks. This doesn't seem like typical tasks that would need the LLMs. Can the approach be evaluated on more complex tasks such as QA?

- Only one model is used for the evaluation. Can the approach be evaluated on more model types (encoder-only, encoder-decoder, and decoder-only) to demonstrate its generalizability?

**Questions:**

see above

---

### Official Review · Reviewer_Lsyx · 2024-11-04

**Soundness:** 1
**Presentation:** 3
**Contribution:** 2
**Rating:** 3
**Confidence:** 3

**Summary:**

The paper introduces **FedQLoRA**, a novel framework designed to fine-tune large language models in a federated learning environment while mitigating quantization bias and data heterogeneity issues. The key contributions are:

1. Identification of **quantization bias** when using adapter-sharing and aggregation in FL settings with LLMs having different levels of quantization.
2. Proposal of **FedQLoRA**, which includes a quantization-aware adapter to separate quantization error from the LoRA adapter. This allows effective aggregation without introducing bias.
3. Introduction of an **iterative version** of FedQLoRA to address data heterogeneity by alternately refining the dynamic quantization-aware adapter and the LoRA adapter, reducing the negative effects of both quantization bias and data heterogeneity.
4. Some experiments validating the superior performance of FedQLoRA compared to existing methods, especially in non-IID data scenarios.

These contributions make FedQLoRA effective for privacy-preserving, resource-efficient model training across heterogeneous devices.

**Strengths:**

- Originality: The paper presents a significant step forward in federated learning for LLMs by identifying the previously unaddressed issue of quantization bias when aggregating LoRA adapters in mixed quantization settings. The proposed FedQLoRA framework introduces an innovative solution by designing a quantization-aware adapter, which effectively separates the quantization error from the local LoRA adapters.
- Quality:  The authors provide a thorough comparison with multiple existing methods, including baselines like LoRA, FFA-LoRA, and H-LoRA, which strengthens the reliability of the findings.
- Clarity: The methodology is clearly articulated, breaking down the components of the FedQLoRA framework into understandable segments, supported by appropriate mathematical formulations.
- Significance: The significance of the paper lies in its potential impact on the broader adoption and efficiency of federated learning for LLMs, especially in environments constrained by heterogeneous hardware and limited communication bandwidth.

**Weaknesses:**

1. Significance of FedQLoRA Needs More Justification: The paper's claim of the significance of FedQLoRA could be further strengthened. If a client has sufficient memory and computational resources to perform quantization-aware learning, it begs the question: why not use the unquantized model instead? This raises concerns regarding the practicality of the proposed framework. Additionally, the authors do not provide a detailed memory consumption footprint for client training. Such details are crucial for understanding the trade-offs between quantization-aware learning and using unquantized models, and for justifying the choice of FedQLoRA in scenarios where resources might be limited.

2. Experimentation with Small-Scale Models: The experiments were conducted on DistilBERT-base-multilingual-cased, which is a relatively lightweight model. This choice does not align well with the context presented in the paper, where clients require parameter-efficient fine-tuning methods like QLoRA to manage limited memory. The use of a smaller model reduces the perceived necessity of quantization and FedQLoRA, thereby diminishing the impact of the proposed solution. A more convincing evaluation would involve a model of larger scale, such as LLaMA3-8B, which would better demonstrate the efficiency gains from using FedQLoRA.

3. Limited Federated Learning Setup: The experimental setup only involves up to 10 clients, which is significantly smaller compared to real-world federated learning applications that typically involve hundreds or thousands of clients. This limitation raises questions about the scalability of the proposed method. The effectiveness of FedQLoRA in handling quantization and heterogeneity biases might vary considerably in larger, more diverse client settings. The authors could benefit from either extending their simulations to include more clients or providing a discussion on the expected scalability and potential bottlenecks when the number of clients is scaled up.

4. Lack of Theoretical Convergence Analysis: The paper lacks a theoretical convergence analysis to support the claims regarding the convergence of the proposed optimization method. While empirical evidence suggests that FedQLoRA and its iterative variant perform well, a formal convergence guarantee is essential to demonstrate the robustness of the proposed approach, especially in a federated learning setting where data distribution and client behavior are often unpredictable. Including such analysis would significantly improve the rigor of the proposed method.

**Questions:**

1. Report Memory Usage and Training Time: For a federated learning setting, reporting the maximum memory usage and end-to-end training time across different clients in the experiments would help in demonstrating the scalability and practical feasibility of the proposed method. This is particularly important given the emphasis on quantization and resource efficiency. Providing such metrics would make the contribution more concrete and would clarify the efficiency gains of FedQLoRA over existing methods.

2. Experiment with Larger, More Popular Models: Consider using a more widely recognized and computationally intensive model, such as LLaMA3-8B, to further validate the proposed approach. Using a larger model that is commonly employed in complicated language tasks would make the experiments more representative and the findings more compelling, especially for audiences interested in deploying federated learning with large models.

3. Expand Number of Clients in Experiments: Increase the number of clients in the experiments to better reflect real-world federated learning scenarios. Current experiments with up to 10 clients are insufficient to demonstrate scalability. Testing the proposed framework on dozens or hundreds of clients would provide better insights into its performance and limitations in more realistic, large-scale settings.

4. Theoretical Convergence Analysis: Including a theoretical convergence analysis would significantly strengthen the claims about the optimization method. Could you provide more formal guarantees or proofs regarding the convergence behavior of the quantization-aware learning approach?

5. Clarify Aggregation Method in Experiments: The paper does not clearly state the aggregation method used in the experimental setup. Providing more details about the aggregation method would help in understanding how the different LoRA adapters are integrated across clients and if any specialized approach contributes to the improvements.

---

### Meta-Review · Area_Chair_6rNQ · 2024-12-21

**Metareview:**

This paper introduces FedQLoRA, a framework designed to fine-tune LLMs in FL while addressing quantization bias and data heterogeneity.  The reviewers appreciated the introduction of the quantization bias concept in federated LLM fine-tuning, the clear presentation of methodology, and the effort for reproducibility. However, several weaknesses limit the quality of this work. Reviewers raised concerns about the practicality of the proposed quantization bias scenario, which appears contrived given that real-world FL systems often use uniform quantization. The lack of experiments on large LLMs, larger-scale FL setups (e.g., with more clients) and comparisons with personalized FL methods further weakens the empirical evaluation. In addition, the theoretical basis of the suggested technique such as convergence guarantees and quantization error approximation is inadequately covered. At last, reviewers pointed out that there is overlap with existing solutions like LoftQ and very little innovation in dealing with heterogeneous quantification techniques.

**Additional Comments On Reviewer Discussion:**

No rebuttal was submitted by the authors. Therefore the concerns remain unaddressed.

---

### Decision · Program_Chairs · 2025-01-22

Reject